# Topography and environmental deficiencies are associated with chikungunya virus exposure in urban informal settlements in Salvador, Brazil

Catherine Tamera Travis[1], Hernán D. Argibay[2,3], Maysa Pellizzaro[2],
Daiana de Oliveira[2,3], Roberta Santana[2], Fabiana Almerinda G. Palma[2], Ricardo Lustosa[4],
Juliet Oliveira Santana[2,3], Fábio Neves Souza[2,3], Yeimi Alexandra Alzate López[2],
Mitermayer G. Reis[3,5,6], Albert I. Ko[3,6], Peter J. Diggle[7], Guilherme S. Ribeiro[3,5],
Michael Begon[8], Federico Costa[2,3,6*⊙], Hussein Khalil[9⊙], Max T. Eyre[10⊙]

**1** Southern Nevada Health District, Las Vegas, Nevada, United States of America, **2** Federal University of Bahia, Collective Health Institute, Salvador, Bahia, Brazil, **3** Gonçalo Moniz Institute, Oswaldo Cruz Foundation, Ministry of Health, Salvador, Bahia, Brazil, **4** Laboratory of Geotechnologies Applied to Sciences, Federal University of Western Bahia, Barra, Bahia, Brazil, **5** Faculty of Medicine of Bahia, Federal University of Bahia, Salvador, Bahia, Brazil, **6** Department of Epidemiology of Microbial Diseases, School of Public Health, Yale University, New Haven, Connecticut, United States of America, **7** Lancaster University, Lancaster Medical School, Lancaster, United Kingdom, **8** Department of Evolution, Ecology and Behaviour, The University of Liverpool, Liverpool, United Kingdom, **9** Department of Wildlife, Fish, and Environmental Studies, Swedish University of Agricultural Sciences, Umeå, Sweden, **10** Environmental Health Group, Faculty of Infectious Tropical Diseases, London School of Hygiene and Tropical Medicine, London, United Kingdom

⊙ Those authors have contributed equally
* fcosta2001@gmail.com

## Abstract

### Background

Chikungunya virus (CHIKV) is an arbovirus with a significant global public health burden. Delineating the specific contributions of individual behaviour, household, natural and built environment to CHIKV transmission is important for reducing risk in urban informal settlements but challenging due to their heterogeneous environments. The aim of this study was to quantify variation in CHIKV seroprevalence between and within four urban communities in a large Brazilian city, and identify the respective contributions of individual, household, and environmental factors for seropositivity.

### Methodology/principal findings

A cross-sectional serological survey was conducted in four low-income communities in Salvador, Brazil in 2018 to collect individual, household and CHIKV IgG serology data for 1318 participants. Fine-scale community mapping of high-risk environmental features and remotely sensed environmental data were used to improve characterisation of the microenvironment close to the household. We categorised risk factors into three domains - individual, household, and environmental and used binomial mixed-effect models to identify associations with CHIKV seropositivity. CHIKV

**Data availability statement:** All R code used for analysis and the de-identified dataset supporting this study are available on the Open Science Framework (OSF) at https://osf.io/4f975/.

**Funding:** The study was funded by the Medical Research Council (UK). Grant number: MR/P024084/1 to MB, Fundação de Amparo à Pesquisa do Estado da Bahia (BR) Grant number: 10206/2015, Wellcome Trust (UK) Grant number:102330/Z/13/Z and National Institutes of Health (US) Grant number:1 R01 TW009504 to FC. MTE was supported by a Reckitt Global Hygiene Institute (RGHI) fellowship. FC was supported by the National Institutes of Health NIH/AID grant numbers F31 AI114245, R01 AI052473, U01 AI088752, R01 TW009504 and R25 TW009338. The funders had no role in study design, data collection and analysis, decision to publish, or preparation of the manuscript. https://www.ukri.org/councils/mrc/ https://www.fapesb.ba.gov.br https://wellcome.org https://www.nih.gov https://rghi.org.

**Competing interests:** I have read the journal's policy and the authors of this manuscript have the following competing interests: AIK has received funding from the National Institutes of Health for work related to Aedes aegypti in Brazil in a cluster-randomized trial to evaluate the efficacy of Wolbachia-infected Aedes aegypti mosquitoes in reducing the incidence of arboviral infection in Brazil (EVITA Dengue) – DMID 17-0111. The authors have no financial relationships with any organisations that might have an interest in the submitted work in the previous three years and no other relationships or activities that could appear to have influenced the submitted work.

seroprevalence was 4.8%, 6.1% and 4.3% in three communities and 22.6% in one community which had a distinct topographical profile. The only individual domain variable associated with seropositivity was male sex (OR 1.67, 95% CI 1.11 - 2.36), but several environmental variables, including living in a house on a steep hillside, at medium to high elevations, and with surface water nearby, were associated with higher seropositivity.

## Conclusions/significance

Our findings indicate that CHIKV exposure risk can vary significantly between nearby communities and at fine spatial scales within communities and is likely to be driven more strongly by the availability of mosquito breeding sites rather than individual exposure patterns. They suggest that environmental deficiencies and topography, a proxy for several environmental processes including the degree of urbanisation and flooding risk, may play an important role in driving risk at both of these scales.

## Author summary

Chikungunya virus, now endemic in Brazil, has seen a concerning rise in cases in urban informal settlements with inadequate water, sanitation, & hygiene infrastructure, infrequent collection of waste, and environmental degradation. Previous research explored various risk factors for chikungunya seropositivity such as demographic characteristics, housing conditions, and environmental determinants—there has been little in-depth study of how these factors interact to influence transmission risk in complex urban settings. Our study addresses this gap by integrating high-resolution environmental data with cross-sectional serology data collected in four informal settlements within Salvador, Brazil to better understand the role that the heterogeneous natural and built environment play in chikungunya virus transmission at fine spatial scales. Our findings indicate that chikungunya virus exposure risk varies significantly between communities and at fine spatial scales within communities, and is likely to be driven more strongly by the availability of mosquito breeding sites rather than human exposure patterns. They suggest that environmental deficiencies and topography, a proxy for environmental processes including the degree of urbanisation and flooding risk, may play an important role in driving risk. Our analysis underscores the importance of addressing ecological and environmental vulnerabilities in urbanising areas to reduce arbovirus transmission.

## Introduction

Chikungunya virus (CHIKV) is an Alphavirus arbovirus that emerged in the 21st century and represents an important public health concern due to its frequent and widespread outbreaks and substantial global burden [1]. Spread by *Aedes albopictus*

and *Aedes aegypti* mosquitoes [2], transmission of CHIKV is particularly intense in tropical urban areas [3] where social, environmental, and climatic conditions result in high availability of breeding sites.

The first CHIKV outbreaks in Brazil occurred in the states of Amapá and Bahia in 2014 and within one year, all other states in Brazil had confirmed cases [4]. Since then, over 1.6 million cases have been reported in Brazil [5]. Due to concurrent outbreaks of CHIKV, dengue (DENV), and Zika viruses (ZIKV), health surveillance systems in Brazil are frequently overwhelmed and there is difficulty measuring the true burden of CHIKV [6–7].

Deprived urban communities, or informal settlements, in Northeast Brazil are transmission hotspots for CHIKV and other arboviruses, and are characterised by high population density, high temperatures, flooding risk, water shortages, and socioeconomic vulnerability [5]. Structural deficiencies, such as the inadequate provision of drainage and sanitation systems, a lack of regular trash collection and inconsistent water supply, are common [8]. These drive mosquito abundance through standing water in the environment, particularly within discarded containers, and the use of on-site water storage systems [8].

Previous research in urban areas of Brazil has highlighted the importance of social vulnerability and environmental and infrastructural deficiencies as drivers of CHIKV infection, identifying a range of risk factors for seropositivity that include socioeconomic status (SES), living conditions and the structural quality of houses and nearby pavement [4,9–11]. These analyses have shown how social, economic, and environmental processes overlap to drive CHIKV transmission, but they also demonstrate the inherent challenges in delineating the specific contributions of individual behaviour and mobility, household structure, peri-domestic environment and the natural and built environment close to and further away from the household.

Accurate characterisation of the processes that drive arboviral infection in humans requires precise measurement of the environmental features that drive mosquito breeding. This is particularly important in urban informal settings which have complex and heterogeneous rural-urban environments that vary significantly over short distances. Consequently, capturing fine-scale variation in environmental features helps to estimate environmental effect alone and in conjunction with individual and household factors. Accurate estimation of the effect environmental features can provide critical data in targeting of vector control within communities. This calls for community-level research that supplements survey variables commonly measured at the household location or municipal level with high-resolution community-mapped environmental data that captures the fine-scale spatial variation in the natural and built environment [12].

The aim of this study was to quantify variation in CHIKV seroprevalence between and within four urban communities in a large Brazilian city and identify individual, household, and environmental risk factors for seropositivity. We aimed to improve characterisation of the peri-domestic environment close to the household by using fine-scale community mapping of high-risk environmental features and remotely sensed environmental data.

## Methods

### Ethics statement

Ethical approval for this study was obtained from the Research Ethics Committee at the Collective Health Institute, Federal University of Bahia, Brazil (permit 041/17 2.245.914.17 2.245.914) and the National Commission for Research Ethics, Brazilian Ministry of Health (CAAE: 68887417.9.0000.5030). All participants involved in the study provided written informed consent before data collection.

### Study site and participant selection

We performed a cross-sectional sero-survey in four communities in the north-western periphery of Salvador: Marechal Rondon (MR), Alto do Cabrito (AC), Nova Constituinte (NC), and Rio Sena (RS). Study areas ranged between 0.07 km² and 0.09 km² (Fig 1). These underserved and low-income communities, also known as *favelas,* slums, or informal settlements, are characterised by poor living and working conditions, and high population density [13]. These communities are

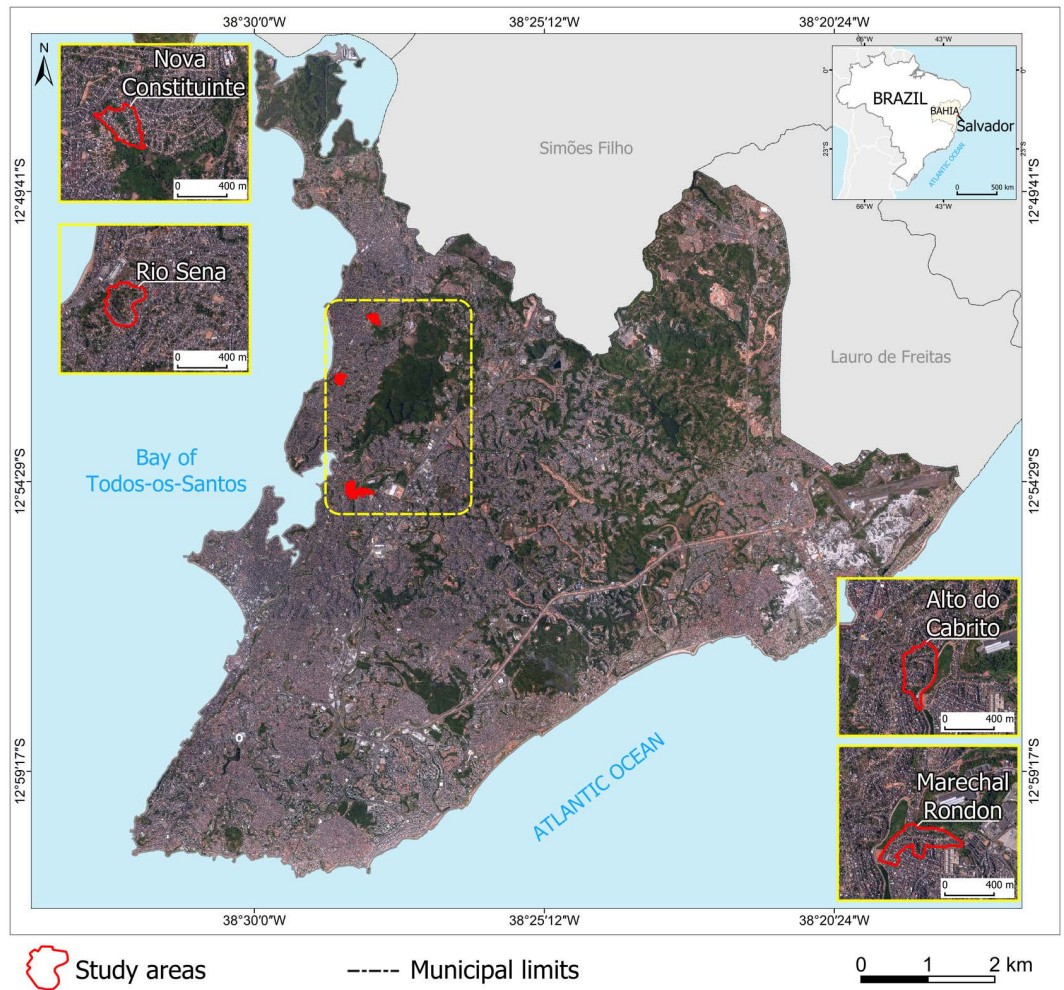

**Fig 1. Study areas Rio Sena, Nova Constituinte, Marechal Rondon, and Alto do Cabrito highlighted in red with the surrounding geography of Salvador, Brazil.** Source: Base image (Salvador/SEFAZ, 2017); Municipal and state limits (IBGE, 2017).

not homogeneous areas and have significant variations in socioeconomic status and environmental features over small distances [4,14].

First, study enumerators conducted a full census of the study areas. All individuals ≥5 years of age who slept ≥3 nights per week within the study areas and had lived there for at least 6 months were invited to participate in the study. Enrolled participants or their legal guardians provided written informed consent for blood sample collection and participation in a survey for potential risk factors for arboviral exposure.

## Household survey

Data was collected between March and October 2018 by trained enumerators using a previously standardised questionnaire [14]. The survey items included questions regarding demographics, income, assets, household features, and peri-domestic environmental factors that were assessed by the data collection team such as the presence of open sewers, flooding, trash accumulation, household trash and water storage, whether the house was located on a hillside, mosquito sightings, and if agents from the Centre for the Control of Zoonoses (CCZ), an agency that provides health education materials and target vector breeding sites, had visited in the past year.

## Serologic evaluation

During household visits, 10 mL of blood was collected from participants and stored between 2°C–8°C, then sent to the Instituto Gonçalo Moniz, Fundação Oswaldo Cruz laboratory. After aliquoting serum from centrifuged samples, the serums were stored at −20°C and tested using a commercial anti-CHIKV IgG ELISA with >90% Relative Sensitivity and Relative Specificity [15]. There was no evidence of significant CHIKV transmission in Salvador before 2015 and these IgG results are consequently representative of recent cases occurring in the period 2015–2018.

We interpreted both the CHIKV IgG results in ordinance with standard manufacturer instructions. CHIKV IgG absorbance/calibrator levels were negative at <0.8, indeterminate at ≥0.8 to <1.1, and positive at ≥1.1 [4]. Samples that first returned indeterminant results were retested and the second result was considered final [4].

## Environmental variables

**Mapped variables.** Trained field teams systematically surveyed all publicly accessible spaces across the four study areas to collect information on the locations of: i) all trash piles covering an area greater than 0.25m$^2$; ii) the open sewer system (including major and minor sewers) mapped as lines. Trash piles and sewers were chosen because of their expected contribution to mosquito breeding risk through rainwater collection in discarded containers and pooling of stagnant water [16]. To create environmental risk variables for use in this analysis, the shortest three-dimensional distance to: i) each trash point and ii) the sewer system was calculated for each household.

**Remotely sensed variables.** Relative elevation was calculated as the difference in elevation above sea level between a given household and the household with the lowest elevation in that community. Land cover rasters were created from fine-scale WorldView-3 satellite images taken in February 2019 (resolution of 0.3m by 0.3m) using the Semi-Automatic Classification Plugin 8.5.0 in QGIS LTR 3.16 [17]. These rasters were classified into four categories: impervious surfaces, exposed earth, vegetation, and water sources. To assign a measure of nearby standing water (a proxy for breeding site risk) to each household, a 20m buffer zone was defined around each household and the proportion of the area that was classified as land covered by water was calculated. While the satellite images were taken one year after the data collection period, these environmental data are minimally variable and are still representative of the exposure window. A Normalised Difference Vegetation Index (NDVI) layer was created using the red and near-infrared bands of the World-View 3 satellite images [18] to capture variations in plant density and health which can affect mosquito habitats [19–20]. Once completed for all communities, the mean NDVI value was generated for a 20m buffer around each household. For all of these remotely-sensed environmental variables, a 20m buffer size was chosen to account for the expected clustering distance of the *Aedes aegypti* mosquito in these highly heterogeneous urban settings [21].

The maps depicting study areas and the spatial distribution of the cases were generated from the analysis of primary data produced by the authors. The data used as a basis for the creation of all the maps, including the location map of the study areas, are not protected by copyright, as they were created by the authors themselves using the free and open source software QGIS, version 3.38.3 [17].The municipal and state boundaries [22], the Orthophoto and the Digital Terrain Model, with Coordinate System: UTM, 24S time zone; Geodetic Reference System: SIRGAS 2000 [23] are also open data and were downloaded from the website of the Brazilian Institute of Geography and Statistics [22] and the City Hall of Salvador [23].

## Statistical analysis

Participant age, education, and income were considered as both continuous and ordered categorical variables. In the first case, we grouped age into ranges of 5–15 years, 16–25, 26–45, 46–65, and ≥66 years to account for the relatively young age sample. Education level was grouped into 0–5 years of education, 6–9 years, 10–12 years, and 13 years or above. Per capita income was calculated as the total monthly household income in BRL including the value of the government assistance programme 'bolsa familia', where applicable, divided by the number of people living in the household. Racial

categories *Preto* (Black) and *Pardo* (mixed ethnicities) were combined following the Brazilian Institute of Geography and Statistics (IBGE) race definition [24].

Generalised additive models (GAMs) were used to assess whether the relationship between each continuous explanatory variable and CHIKV seropositivity was linear. As there was evidence against linearity for relative elevation and distance to trash, piecewise linear splines (also known as 'broken stick splines') were used with a single knot placed at 17m for relative elevation and 50m for distance to trash (S1 and S2 Figs). This gives two regression estimates for each variable, consisting of the gradient (i.e., change in log-odds of seropositivity per unit increase) for the first interval (0m to the knot value), and a second value which is the gradient for the second interval (knot value to largest observed value) and provides a measure of how absolute risk changes after the knot value.

Variables were grouped into 3 domains: 1. Individual variables - sociodemographic factors and exposure-related behaviours, such as age, race, income, education, and behaviours that may act as proxies for general skin covering and higher potential vector exposure (e.g., walking barefoot outdoors); 2. Household variables that relate to factors inside or directly surrounding the home of participants, such as housing construction type and the functionality of household features, e.g., served with running water; 3. Environmental variables which included garbage collection service, distance to trash locations and sewers, relative elevation, NDVI, and water land cover. A full description of survey variables is provided in S1 Table.

Univariable analyses were performed using a mixed-effects logistic regression with random effects to account for clustering at the household level. Variable selection for the multivariable model was conducted for each domain separately. Models were fitted for all combinations of variables within each domain and ranked according to their small-sample corrected Akaike Information Criterion (AICc) value. The most parsimonious model was chosen for each domain, defined as the model with the fewest variables within 2 AICc of the lowest value. The variables selected from each domain were then carried into a final round of variable selection in which the same process was repeated for all of these variables to determine which of them would be included in the final model. Age, sex, and community were considered *a priori* confounders and were included in all models.

We assessed multicollinearity among the independent variables using the Variance Inflation Factor (VIF). A VIF value greater than 10 was considered indicative of significant multicollinearity, but this was not found for any variables or models.

To check for spatial autocorrelation in the residuals of the final model, the package PrevMap [25] as used. A variogram is provided in S3 Fig, which showed no evidence of residual correlation and no need for a geostatistical model to be fit.

We performed the data analysis using R version 3.6.3 [26] and packages tidyverse [27], gmodels [28], and lme4 [29].

## Results

### Description of study population

In the four communities, there were 2590 eligible individuals, of which 1318 (50.9%) consented to join the study and provided a blood sample for testing, and 1316 (99.8%) had conclusive serological results. A full description of the study population stratified by explanatory variables is provided in Table 1 with subgroup seroprevalences. Overall, study participants of all areas were relatively young, with median ages ranging from 26 in RS to 38 in MR (S2 Table), and an overall median age of 33 (IQR 19–48 years). The study population had more women than men (57.3%), was primarily Black or of mixed race (90.9%), and most participants had a relatively low monthly income per capita with a median of around 234 BRL (IQR 12–460 BRL equivalent to 3.3-126.32 USD/month). Three out of the four areas had similar mean absolute household elevations: MR had a mean of 49.62m (IQR 44.0-55.4m) above sea level, AC was 54.74m (IQR 45.0-61.76m), RS was 64.65m (IQR 57.54- 73.00), but NC was far lower than other communities at 8.54m (IQR 4.83- 12.21m) (S4 Fig). This difference was apparent in the proportion of participants in each community that reported living in a household on a hillside, with only 3.3% of participants reporting this in NC compared to 27.6%, 20.8% and 59.1% in MR, AC and RS, respectively (S3 Table).

**Table 1. CHIKV seroprevalence, determined by detection of IgG, stratified by demographic, behavioural, and environmental factors (n = 1316).**

| Variable | | Number of participants (%) | Seropositive (%) |
|---|---|---|---|
| Individual Variables | | | |
| Sex | Female | 754 (57.3) | 57 (7.6) |
| | Male | 562 (42.7) | 64 (11.5) |
| Age | 5–15 | 233 (17.7) | 22 (9.4) |
| | 16–25 | 262 (19.9) | 30 (11.5) |
| | 26–45 | 442 (33.6) | 44 (10.0) |
| | 46–65 | 297 (22.6) | 19 (6.4) |
| | >66 | 73 (5.6) | 6 (8.2) |
| Race | White | 80 (6.1) | 8 (10.0) |
| | Black/ mixed race | 1196 (90.9) | 109 (9.1) |
| | Asian | 28 (2.1) | 2 (7.1) |
| | Indigenous | 12 (0.9) | 2 (16.7) |
| Education (years) | 0–5 | 388 (29.5) | 35 (9.0) |
| | 6–9 | 396 (30.1) | 37 (9.3) |
| | 10–12 | 467 (35.5) | 47 (10.1) |
| | Upper levels (>13) | 36 (2.7) | 2 (5.6) |
| Income per capita (BRL) | 0–99 | 403 (30.6) | 47 (11.7) |
| | 100–300 | 341 (25.9) | 34 (10.0) |
| | 300–500 | 311 (23.6) | 27 (8.7) |
| | 500–1000 | 195 (14.8) | 12 (6.2) |
| | >1001 | 37 (2.8) | 1 (2.7) |
| Walk through sewage | No | 1040 (79.0) | 95 (9.1) |
| | Yes | 276 (21.0) | 26 (9.4) |
| Walk through floodwater | No | 822 (62.5) | 81 (9.9) |
| | Yes | 494 (37.5) | 40 (8.1) |
| Walk through mud | No | 825 (62.7) | 95 (11.5) |
| | Yes | 490 (37.2) | 26 (5.3) |
| Walk outside barefoot | No | 824 (62.6) | 70 (8.5) |
| | Yes | 492 (37.4) | 51 (10.4) |
| Boot access | No | 1085 (82.4) | 96 (8.8) |
| | Own boots | 210 (16.0) | 23 (11.0) |
| | Can borrow boots | 19 (1.4) | 2 (10.5) |
| Household Variables | | | |
| Lacking water within 30 days | No | 704 (53.5) | 55 (7.8) |
| | Yes | 611 (46.5) | 66 (10.8) |
| Running water in home | No | 31 (2.4) | 4 (12.9) |
| | Yes | 1284 (97.6) | 117 (9.1) |
| Paved home entry | No | 316 (24.0) | 19 (6.0) |
| | Yes | 999 (75.9) | 102 (10.2) |
| Wall material | Covered concrete or brick | 1161 (88.2) | 104 (9.0) |
| | Other | 154 (11.7) | 12 (7.8) |
| House with yard | No | 442 (33.6) | 37 (8.4) |
| | Yes | 873 (66.3) | 84 (9.6) |
| House on hillside | No | 958 (72.8) | 92 (9.6) |
| | Yes | 357 (27.1) | 29 (8.1) |

*(Continued)*

**Table 1.** (Continued)

| Variable | | Number of participants (%) | Seropositive (%) |
|---|---|---|---|
| Mosquito in home | No | 442 (33.6) | 43 (9.7) |
| | Yes | 873 (66.3) | 78 (8.9) |
| Water storage in home | No | 336 (25.5) | 28 (8.3) |
| | Yes | 979 (74.4) | 93 (9.5) |
| Rainwater accumulates in home | No | 1141 (86.7) | 104 (9.1) |
| | Yes | 174 (13.2) | 14 (8.1) |
| CCZ visitation | Within past year | 965 (73.3) | 104 (10.8) |
| | Over 1 year/never | 350 (26.6) | 17 (4.9) |
| Environmental Variables | | | |
| Community | Marechal Rondon | 337 (25.6) | 16 (4.8) |
| | Alto do Cabrito | 376 (28.6) | 23 (6.1) |
| | Nova Constituinte | 305 (23.2) | 69 (22.6) |
| | Rio Sena | 298 (22.6) | 13 (4.4) |
| Open sewer near house | No | 886 (67.3) | 78 (8.8) |
| | Yes | 427 (32.5) | 43 (10.1) |
| Streetlights | No | 57 (4.3) | 11 (19.3) |
| | Yes | 1256 (95.4) | 110 (8.8) |
| Trash storage in home | In plastic bags only | 1170 (88.9) | 112 (9.6) |
| | In containers with lids only | 25 (1.9) | 0 (0.0) |
| | Both in plastic bags and containers with lids | 117 (8.9) | 9 (7.7) |
| | Other | 3 (0.2) | 0 (0.0) |
| Garbage collection | No | 225 (17.1) | 12 (5.3) |
| | Yes | 1090 (82.8) | 109 (10.0) |

## Seroprevalence

Of the 1318 participants who provided a blood sample, 1316 (99.7%) had conclusive serological results with 121 participants (9.2%) testing positive for CHIKV-specific antibodies. CHIKV seroprevalence across the four study areas was 6.1% (n = 23/376) in AC, 4.8% (n = 16/337) in MR, 22.6% (n = 69/305) in NC, and 4.3% (n = 13/298) in RS. Households with seropositive individuals are marked in Fig 2 with relative elevation shown for the four study areas. Nova Constituinte was the flattest study area (with areas in the range of 0-30m), Marechal Rondon and Alto do Cabrito had more mixed elevation areas (0-50m), and Rio Sena was the steepest study area with considerable variation in elevation (0-75m). The majority of positive households in most study areas were found just above the lowest relative elevation areas.

## Age and sex stratified seroprevalence

The seroprevalence in men of 11.4% (n = 64/562) was higher than in women, 7.6% (n = 57/754). There was no clear pattern in seroprevalence across age groups within male and female participants (Fig 3). Within males, those aged 16–25 had the highest seroprevalence, whereas in females seroprevalence was highest in the 26–45 age group.

## Univariable analysis

In the univariable analysis (Table 2), sex was the only individual-level variable that was associated with CHIKV seropositivity, with male participants more likely to test positive for CHIKV than female participants (OR 1.77, 95% CI 1.09 − 2.88).

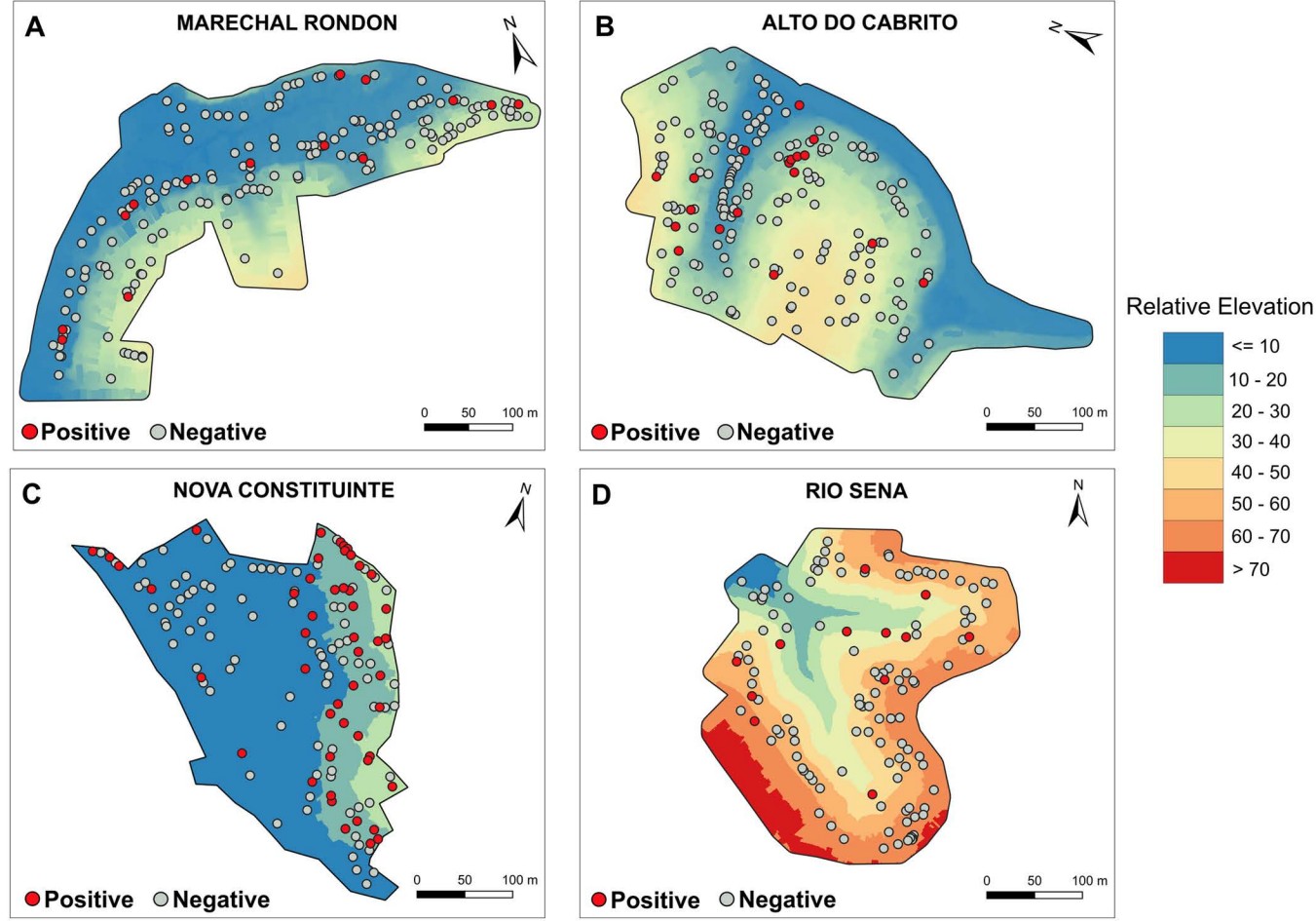

**Fig 2. Maps of relative elevation (m) and household seropositivity (red circle – at least one participant was seropositive; grey circle – all participants negative) shown for each study area. A)** Marechal Rondon, **B)** Alto do Cabrito, **C)** Nova Constituinte, **D)** Rio Sena.". Source: Digital Terrain Model (Salvador/SEFAZ, 2017).

In the environmental domain, participants living in community NC had 10 times the odds of being seropositive relative to those living in MR (OR 10.33, 95% CI 4.24 – 25.22) and participants living in households further away from sewers had slightly higher odds of being seropositive (OR 1.05 per 10m, 95% CI 1.00 – 1.10). The odds of being seropositive increased for each additional metre of household elevation relative to the bottom of each community up to a relative elevation of 17m (elevation 0-17m: OR 1.13 per 1m, 95% CI 1.07 – 1.22), above which there was a negative association (elevation > 17m spline: OR 0.93 per 1m, 95% CI 0.88 – 0.98) modelled as a linear piecewise spline with a single knot at 17m. See the GAM plot of this relationship in S2 Fig for a visualisation of this relationship.

In the household domain, seropositivity was inversely associated with household per-capita income (OR 0.83 per 500 BRL, 95% CI 0.70 – 0.98). Participants living in households located on a hillside also had a higher odds of being seropositive (OR 2.76, 95% CI 1.26 – 6.02). The finding that living on a hillside was associated with higher seropositivity contrasted with the trend in seroprevalences described in Table 1 which found a higher seroprevalence in non-hillside households of 9.6% compared to 8.1% in hillside households. A sub-analysis was conducted to examine the number of participants living in hillside and non-hillside households and their seroprevalence (see S3 Table). The difference was

PLOS Neglected Tropical Diseases

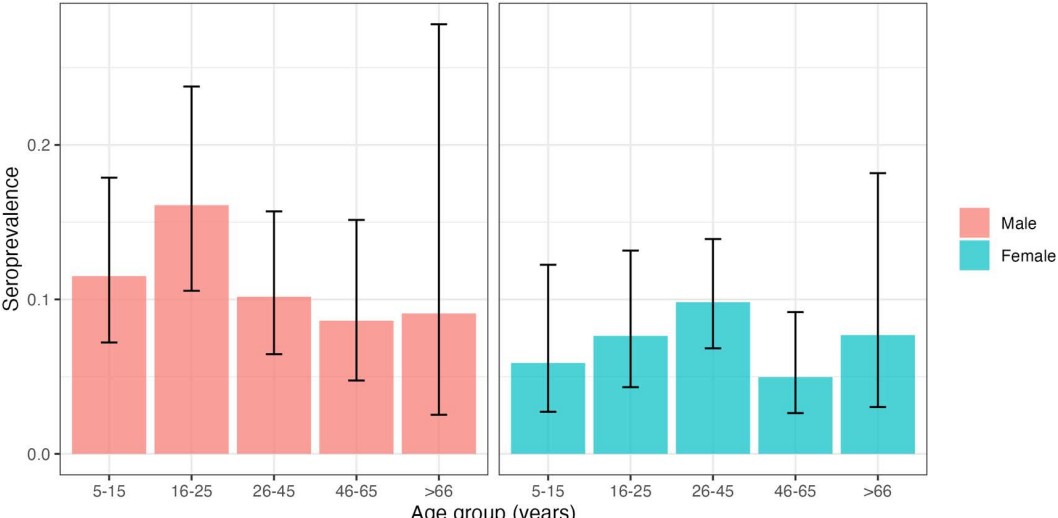

**Fig 3. Age and sex-stratified seroprevalence.** Participant seropositivity by age and sex from all four combined study areas.

found to be driven by confounding with the community variable which was adjusted for in the univariable model due to differences in the topography in NC. This was caused by the significantly higher seroprevalence in both hillside (40.0%) and non-hillside households (22.0%) in the NC community compared to other communities. Seroprevalence in the other three areas ranged from 5.7-8.9% in hillside households and 2.5-5.4% in non-hillside households. Within every community, the seroprevalence was higher in hillside households. Consequently, the higher seroprevalence in non-hillside compared to hillside households in the crude estimates in Table 1 were driven by the fact that in NC there were only 10 individuals living in hillside households compared to 295 in non-hillside households, resulting in a significant upweighting of the seroprevalence of non-hillside households across the study.

## Multivariable analysis- model selection

Of the 1316 participants, 1287 (97%) had complete responses for all variables and were included in the multivariable analysis. Each stage of the domain-specific model selection process is shown in S4-S6 Tables. Only the three *a priori* confounders (community, sex, and age) were maintained for the individual model. Water storage in the home, house on a hillside, and CCZ visitation were selected from the household model and open sewer, water land cover, and relative elevation were selected for the environmental model. The environmental domain model provided a better model fit than the individual and household domain models, as shown by its lower AICc of 706.6. All variables from each of the three domain models were selected in the final multi-domain multivariable model except for water storage in the home and whether there was an open sewer within 10m of the house. Full final model selection can be found in S7 Table.

## Multivariable risk factors

In the final multivariable model (Table 3), male participants had a higher estimated odds of being seropositive (OR 1.72, 95% CI 1.12 – 2.65), but there were no clear differences by age group. Participants living in the NC community were much more likely to be seropositive (OR 8.66, 95% CI 4.06 – 18.49 relative to MR community) than participants living in any of the other three communities, all of which had a similar odds of seropositivity. The position of a participant's household within each community was important for determining exposure risk. Participants living in a house on a hillside had increased odds of seropositivity (OR 2.16, 95% CI 1.15 – 4.03) and those living at the lowest elevation in each community

**Table 2. Univariable mixed-effects logistic regression model estimates for the probability of a participant being seropositive adjusting for age, sex and community. Each explanatory variable of interest is grouped by domain with adjusted odds ratios (aOR), 95% confidence intervals (95%CI) and p-values shown.**

| Variable | | aOR | CI 95% | p value |
|---|---|---|---|---|
| Individual Variables | | | | |
| Sex | Female | | | |
| | Male | 1.78 | 1.10 – 2.89 | 0.02 |
| Age | 5–15 | | | |
| | 16–25 | 1.79 | 0.82 – 3.79 | 0.18 |
| | 26–45 | 1.52 | 0.75 – 3.09 | 0.31 |
| | 46–65 | 0.91 | 0.39 – 2.14 | 0.75 |
| | >65 | 1.47 | 0.42 – 5.17 | 0.59 |
| Race | White | | | |
| | Black/ mixed race | 0.76 | 0.28 – 2.11 | 0.60 |
| | Asian | 0.80 | 0.11 – 5.86 | 0.86 |
| | Indigenous | 2.15 | 0.20 – 23.15 | 0.53 |
| Education (years) | 0–5 | | | |
| | 6–9 | 0.96 | 0.49 – 1.89 | 0.87 |
| | 10–12 | 1.01 | 0.48 – 2.12 | 0.98 |
| | Upper Levels (>12) | 0.60 | 0.10 – 3.61 | 0.56 |
| Walk through sewage | No | | | |
| | Yes | 1.18 | 0.65 – 2.15 | 0.59 |
| Walk through floodwater | No | | | |
| | Yes | 0.69 | 0.40 – 1.17 | 0.17 |
| Walk through mud | No | | | |
| | Yes | 0.92 | 0.55 – 1.57 | 0.78 |
| Walk outside barefoot | No | | | |
| | Yes | 1.42 | 0.80 – 2.52 | 0.23 |
| Household Variables | | | | |
| Income per capita (BRL); continuous (per 500BRL increase) | | 0.83 | 0.70 – 0.98 | 0.03 |
| Income per capita (BRL); categorical | 0–99 | | | |
| | 100–299 | 0.83 | 0.40 – 1.81 | 0.67 |
| | 300–499 | 0.77 | 0.36 – 1.68 | 0.52 |
| | 500–1000 | 0.40 | 0.14 – 1.10 | 0.08 |
| | >1000 | 0.12 | 0.01 – 1.56 | 0.11 |
| House on hillside | No | | | |
| | Yes | 2.76 | 1.26 – 6.02 | 0.01 |
| Lacking water within 30 days | No | | | |
| | Yes | 0.86 | 0.46 – 1.60 | 0.64 |
| Running water in home | No | | | |
| | Yes | 1.05 | 0.16 – 6.73 | 0.96 |
| Paved home entry | No | | | |
| | Yes | 1.64 | 0.78 – 3.43 | 0.19 |
| Wall material | Covered concrete or brick | | | |
| | Other | 1.17 | 0.46 – 3.00 | 0.74 |
| House with yard | No | | | |
| | Yes | 0.80 | 0.43 – 1.51 | 0.50 |

*(Continued)*

**Table 2.** (Continued)

| Variable | | aOR | CI 95% | p value |
|---|---|---|---|---|
| Mosquito in home | No | | | |
| | Yes | 1.14 | 0.61 – 2.14 | 0.68 |
| Water storage in home | No | | | |
| | Yes | 1.87 | 0.92 – 3.83 | 0.08 |
| Rainwater accumulates in home | No | | | |
| | Yes | 0.96 | 0.40 – 2.30 | 0.93 |
| CCZ visitation | Within past year | | | |
| | Over 1 year/never | 0.50 | 0.23 – 1.08 | 0.08 |
| Environmental Variables | | | | |
| Community | Marechal Rondon (MR) | | | |
| | Alto do Cabrito (AC) | 1.30 | 0.54 – 3.14 | 0.55 |
| | Nova Constituinte (NC) | 10.20 | 4.21 – 24.80 | <0.001 |
| | Rio Sena (RS) | 0.90 | 0.33 – 2.42 | 0.83 |
| Open sewer near house | No | | | |
| | Yes | 1.29 | 0.69 – 2.42 | 0.42 |
| Streetlights | No | | | |
| | Yes | 0.33 | 0.10 – 1.08 | 0.07 |
| Garbage collection | No | | | |
| | Yes | 0.97 | 0.39 – 2.4 | 0.95 |
| Water land cover within 20m from home (per 1%) | | 1.01 | 0.99 – 1.03 | 0.40 |
| Normalised Difference Vegetation Index (per unit) | | 3.82 | 0.17 – 87.81 | 0.40 |
| Relative elevation (per 1m)[1] | For houses located 0–17 | 1.13 | 1.07 – 1.22 | <0.001 |
| | For houses located >17 | 0.93 | 0.88 – 0.98 | <0.001 |
| Distance to trash (per 10m)[1] | For houses located 0–50 | 1.09 | 0.87 – 1.36 | 0.44 |
| | For houses located >50 | 0.61 | 0.33 – 1.11 | 0.11 |
| Distance to sewer (per 10m) | | 1.05 | 1.00 – 1.10 | 0.04 |

[1]The effect of relative elevation and distance to trash are modelled as linear piecewise splines with a single knot at an elevation of 17m and 50m away from the nearest trash pile, respectively. This was informed by the relationship described by GAMs (S1c and S2c Figs).

were estimated to have the lowest odds of seropositivity. The odds of seropositivity positively increased for every additional metre of relative elevation from 0m up to 17m (OR 1.15 per 1m, 95% CI 1.07 – 1.23). Above 17m there was a change in this relationship, with an odds ratio of 0.93 (per 1m, 95% CI 0.88 – 0.98), showing a slight decrease in odds with increase elevation above this point (as was seen in the GAM plot in S2c Fig). The proportion of land cover classified as water within a 20m buffer around the household was positively associated with seropositivity (OR 1.01 per 1%, 95% CI 1.00 – 1.02), but CCZ visitation (OR 0.55, 95% CI 0.29 – 1.04) which was selected in the final model was not associated with seropositivity.

## Discussion

In this multi-community cross-sectional study, we found an overall CHIKV seroprevalence of 9.2% across the four urban study areas although the community with lowest average elevation and flattest topography, NC, had a significantly higher seroprevalence of 22.6%. We explored individual, household, and environmental risk factors and found that participant sex, environmental factors such as the elevation and position of a household on a hillside, and the amount of surface

**Table 3. Multivariable mixed-effects logistic regression final model estimates for the probability of a participant being seropositive with odds ratios (OR), 95% confidence intervals (95%CI) and p-values shown.**

| Variable | | OR | CI 95% | p value |
|---|---|---|---|---|
| Sex | Female | | | |
| | Male | 1.72 | 1.12 – 2.65 | 0.01 |
| Age | 5–15 | | | |
| | 16–25 | 1.84 | 0.93 – 3.64 | 0.08 |
| | 26–45 | 1.48 | 0.79 – 2.77 | 0.22 |
| | 46–65 | 0.99 | 0.47 – 2.08 | 0.97 |
| | 66+ | 1.49 | 0.50 – 4.44 | 0.47 |
| Community | Marechal Rondon (MR) | | | |
| | Alto do Cabrito (AC) | 1.00 | 0.46 – 2.18 | 1.00 |
| | Nova Constituinte (NC) | 8.66 | 4.06 – 18.49 | <0.001 |
| | Rio Sena (RS) | 1.18 | 0.33 – 4.19 | 0.80 |
| CCZ visitation | Within past year | | | |
| | Over 1 year/never | 0.55 | 0.29 – 1.04 | 0.07 |
| House on hillside | No | | | |
| | Yes | 2.16 | 1.15 – 4.03 | 0.02 |
| Water land cover (per 1%) | | 1.01 | 1.00 – 1.02 | 0.04 |
| Relative elevation (per 1m)[1] | For houses located 0-17m | 1.15 | 1.07 – 1.23 | <0.001 |
| | For houses located > 17m | 0.93 | 0.88 – 0.98 | <0.001 |

[1]The effect of relative elevation is modelled as a linear piecewise spline with a single knot at an elevation of 17m above sea level. This was informed by the relationship described by Generalised Additive Modelling (S2c Fig).

water near the household were most strongly associated with CHIKV seropositivity. Despite the inclusion of fine-scale measurement of environmental features through the use of high-resolution mapped variables, only two factors related to the environmental context were included in the final model, namely relative elevation and water land cover.

The finding that NC community had a seroprevalence around four times higher than the other three communities presents further evidence that the intensity of CHIKV transmission can vary significantly over relatively small distances between communities and nearby cities. NC is located approximately 2km from RS and 5km from MR and AC. The seroprevalence estimates for the other three communities were lower than the 12% found in a seroprevalence study conducted in other nearby communities in Salvador just over a year earlier (2016–2017) [4], while the high seroprevalence in NC was similar to the 22% found in a recent study (2017–2017) in a nearby city [11]. CHIKV transmission has been characterised by local outbreak dynamics [9,30], and because of the heightened prevalence of CHIKV-specific antibodies in the residents of NC compared to what was observed in the other study areas it is possible that this community experienced a local outbreak recently.

From our findings in the environmental domain, the higher seroprevalence in NC may also be explained by the community's general topography and elevation relative to nearby communities. In contrast to the other study areas which are located at higher elevations within steep and narrow valleys, NC is located on a planar surface that is situated at a relatively low elevation above sea level. The mean household elevation relative to the lowest household in each community was 9.8m for MR, 14.7m for AC, 44.6 for RS, and 5.7m for NC. This difference in community topography places NC at higher risk of flooding due to the large hydrological catchment area above it, and at a higher risk of standing water forming because of its flat profile limiting water runoff [14,31]. This is consistent with the finding that more surface water near the household was associated with an increased risk of seropositivity. Consequently, while all four study areas have inadequate drainage systems which lead to flooding, NC may be more likely to be most heavily affected by periods of

heavy rainfall and flooding, resulting in a higher availability of short and long-term breeding habitat [32–33] and possibly increased nutrient provision to existing larval containers [34]. Conversely, RS had the highest mean elevation and the lowest prevalence of CHIKV (although it was the most similar to AC and MR) within this study. Understanding the mechanisms by which these topographical characteristics may impact transmission may be useful for developing and targeting community-based vector control and environmental interventions.

The study findings also suggested that CHIKV exposure risk was driven by fine-scale variation in the environment within each community. We identified a non-linear association with household elevation, with seropositivity lowest in participants living at the lowest elevation areas of the communities and found to increase with elevation up to 17 metres of relative elevation after which it decreased slightly. This pattern could also be seen in the maps of household locations with seropositivity marked in Fig 1. In this urban informal setting, living closer to the bottom of the valley is indicative of greater social marginalisation, with households located in these areas found to have lower socioeconomic status [35], poorer water, sanitation, and hygiene (WASH) infrastructure provision, low-quality housing, inadequate trash disposal and a greater risk of exposure to contaminated floodwater [36]. One explanation for why this may be protective for CHIKV exposure may be that these low-elevation areas are also the least urbanised, consisting of vegetation and soil land cover rather than concrete paving and with a lower population density than higher elevation areas, and are at high risk of severe and regular flooding with highly contaminated water (due to open sewers and other sources of environmental contamination). This is consistent with the clustering of cases in NC, a high flood-risk community, which were predominantly found at 7-15m relative elevation, above the bottom of the community. This suggests that there may be a relationship with elevation whereby at medium elevations cleaner water can pool in plastic containers, trash, and paved areas and at lowest elevations there are fewer possible breeding sites due to lower levels of urbanisation and the flooding frequency being so high, or flood water so contaminated with organic material, that it flushes out or contaminates viable breeding sites. The inverse relationship above an elevation of 17m can be explained by better infrastructure and social conditions with better access for trash collection that is commonly found at the top of the valleys which these communities sit within, resulting in fewer breeding sites.

Within the household domain, the finding that houses located on hillsides had a higher risk of seropositivity highlights the interplay between topography and social marginalisation as a driver of risk and is consistent with this relative elevation hypothesis. In rapidly urbanising areas like Salvador, informal housing for the most marginalised populations is common on steep hillsides that are not typically considered suitable for building. Households on hillsides can be found at all elevations in these highly urbanised settings but are generally of much lower low socioeconomic status and are more poorly served by basic urban services than other households at these elevation levels [37]. This makes them more likely to have breeding sites than other households not on hillsides at the same elevation levels. This is consistent with the finding that households who had been visited by CCZ agents more recently had a higher odds of seropositivity as they prioritise the most vulnerable households.

In terms of the individual domain, we found that men were more likely to be CHIKV seropositive than women, and that more specifically, young men of early working age (16–25 years old) had the highest seroprevalence by age and sex. This may be in part a result of men travelling longer average distances within the community during the day than women, as was found in a previous study in Salvador [38]. In communities at high risk for CHIKV, there is risk of mosquito exposure close to home, but travelling long distances around communities in areas with high environmental risk or working outside can further increase exposure to mosquitoes away from the household and may be an explanation for greater seroprevalence in men compared to women. Two previous studies in Salvador found similar seroprevalences between sexes [4,28], although Anjos et al. 2023 found that among men aged 15–29 years the seroprevalence in men was significantly higher than women (18.1% vs 7.4%). A 2021 study of CHIKV in city near to the four study areas found a higher seroprevalence of CHIKV in women than men, however, the authors noted that having high rates of refusal in male participants could have biased this finding [11].

Environmentally transmitted diseases, such as leptospirosis, that have been studied in these areas extensively depend on both individual and contextual risk factors [14] including resident knowledge, attitudes, and practices [39]. However, for mosquito-borne pathogens, transmission depends more on environmental factors promoting vector breeding and presence rather than individual behaviours in low-income urban settings. Our results show that there are significant risk gradients for CHIKV exposure between and within communities, which are likely to be driven by the availability of mosquito breeding sites rather than human exposure patterns. This availability appears to be driven by topography and environmental deficiencies, including flooding risk, infrastructure provision and the type of environment close to the household, all of which are challenging to measure at high spatial resolutions. While elevation may act as a proxy for these environmental processes in this urban setting, future studies should aim to improve on the suite of high-resolution environmental variables used in this study. In particular, our results show that capturing long-term water bodies is insufficient and there is a need for measuring flooding risk more directly through measures such as the topographic wetness index (TWI) or through identification of short-term breeding habitat using higher temporal resolution satellite imaging after heavy rainfall.

A limitation of this study is its cross-sectional study design. As CHIKV-specific IgG antibodies can remain for over 12 months [40], exposure to infected mosquitoes could have taken place at a time when the environment, household, or individual risk factors we measured may not be fully representative of the potential exposure period. Future longitudinal studies are therefore needed to measure CHIKV incidence and identify fine-scale risk factors and gradients at the time of exposure, and will also enable the interplay between meteorological factors, flooding, the environment and CHIKV transmission to be studied at high temporal and spatial resolutions.

This study has shown the importance of identifying environmental risk factors for CHIKV transmission at both community and within-community scales, and highlighted the challenges associated with accurately measuring these environmental processes in highly heterogeneous urban informal settlements. These findings demonstrate how increasingly urbanised and marginalised populations globally are forced to live in structurally and topographically unsafe environments that place them at high risk of arboviral infection.

## Supporting information

**S1 Fig. Generalised Additive Model graphs for continuous variables a) per capita income, b) education, c) distance to trash, and d) age.**
(TIFF)

**S2 Fig. Generalised Additive Model graphs for continuous variables a) elevation above sea level, b) sewer distance, and c) relative elevation.**
(TIFF)

**S3 Fig. Empirical variogram of the final model with semivariance shown against spatial distance in metres.**
(TIFF)

**S4 Fig. Box plot of community height above sea level.**
(TIFF)

**S1 Table. Responses to these variables are based on information provided from interviews the field team asked the head of each surveyed household.** All questions regarding environmental factors were referring to a distance of 10 metres from the house.
(XLSX)

**S2 Table. Median Age of Community Residents.**
(XLSX)

**S3 Table. Comparison of number of participants and seroprevalence for participants living in households on hillsides and not living on hillsides stratified by community.**
(XLSX)

**S4 Table. Five lowest AIC selection table for individual domain variables.**
(XLSX)

**S5 Table. Five lowest AIC selection table for household domain variables.**
(XLSX)

**S6 Table. Five lowest AIC selection table for environmental domain variables.**
(XLSX)

**S7 Table. Final model selection.**
(XLSX)

## Acknowledgments

We thank the residents and community leaders of the neighbourhoods of Marechal Rondon, Alto do Cabrito, Nova Constuinte, and Rio Sena for their support and participation in this study.

## Author contributions

**Conceptualization:** Catherine Tamera Travis, Fábio Neves Souza, Yeimi Alexandra Alzate López, Mitermayer Galvão Reis, Albert I. Ko, Guilherme S. Ribeiro, Michael Begon, Federico Costa, Hussein Khalil.

**Data curation:** Hernán D. Argibay.

**Formal analysis:** Catherine Tamera Travis, Roberta Santana, Juliet Oliveira Santana.

**Funding acquisition:** Yeimi Alexandra Alzate López, Mitermayer Galvão Reis, Albert I. Ko, Peter J. Diggle, Michael Begon, Federico Costa, Hussein Khalil, Max T. Eyre.

**Investigation:** Hernán D. Argibay, Maysa Pellizzaro, Daiana de Oliveira, Roberta Santana, Fabiana Almerinda G. Palma, Ricardo Lustosa.

**Methodology:** Catherine Tamera Travis, Daiana de Oliveira, Peter J. Diggle, Max T. Eyre.

**Project administration:** Ricardo Lustosa, Fábio Neves Souza, Yeimi Alexandra Alzate López, Michael Begon, Federico Costa, Hussein Khalil, Max T. Eyre.

**Resources:** Mitermayer Galvão Reis, Albert I. Ko, Guilherme S. Ribeiro.

**Supervision:** Hussein Khalil, Max T. Eyre.

**Visualization:** Juliet Oliveira Santana.

**Writing – original draft:** Catherine Tamera Travis.

**Writing – review & editing:** Hernán D. Argibay, Daiana de Oliveira, Ricardo Lustosa, Mitermayer Galvão Reis, Albert I. Ko, Peter J. Diggle, Guilherme S. Ribeiro, Michael Begon, Federico Costa, Hussein Khalil, Max T. Eyre.

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
