## [Decision Letter · Decision Letter 0]

27 Apr 2025

Topography and environmental deficiencies are associated with chikungunya virus exposure in urban informal settlements in Salvador, Brazil

Dear Dr. Costa,

Thank you for submitting your manuscript to PLOS Neglected Tropical Diseases. After careful consideration, we feel that it has merit but does not fully meet PLOS Neglected Tropical Diseases's publication criteria as it currently stands. Therefore, we invite you to submit a revised version of the manuscript that addresses the points raised during the review process.

Please submit your revised manuscript within 60 days Jun 26 2025 11:59PM. If you will need more time than this to complete your revisions, please reply to this message or contact the journal office at plosntds@plos.org. Please include the following items when submitting your revised manuscript:

We look forward to receiving your revised manuscript.

Kind regards,

Elvina Viennet

Section Editor

Shaden Kamhawi

co-Editor-in-Chief

Paul Brindley

co-Editor-in-Chief

**Journal Requirements:**

At this stage, the following Authors/Authors require contributions: Catherine Travis, Hernán Argibay, Maysa Pellizzaro, Daiana Santos de Oliveira, Roberta Santana, Fabiana Almerinda G. Palma, Ricardo Lustosa, Fábio Neves Souza, Yeimi Alexandra Alzate López, Mitermayer Galvão Reis, Albert I. Ko, Peter J. Diggle, Guilherme S. Ribeiro, Michael Begon, Federico Costa, Hussein Khalil, and Max Eyre. Please ensure that the full contributions of each author are acknowledged in the "Add/Edit/Remove Authors" section of our submission form.

4) We notice that your supplementary Figures, and Tables are included in the manuscript file. Please remove them and upload them with the file type 'Supporting Information'. Please ensure that each Supporting Information file has a legend listed in the manuscript after the references list.

Potential Copyright Issues:

i) Figure 1. Please (a) provide a direct link to the base layer of the map (i.e., the country or region border shape) and ensure this is also included in the figure legend; and (b) provide a link to the terms of use / license information for the base layer image or shapefile. We cannot publish proprietary or copyrighted maps (e.g. Google Maps, Mapquest) and the terms of use for your map base layer must be compatible with our CC BY 4.0 license.

**Comments to the Authors:**

**Please note that one of the reviews is uploaded as an attachment.**

**Reviewers' Comments:**

Reviewer's Responses to Questions

**Key Review Criteria Required for Acceptance?**

**Methods**

-Are the objectives of the study clearly articulated with a clear testable hypothesis stated?

-Is the study design appropriate to address the stated objectives?

-Is the population clearly described and appropriate for the hypothesis being tested?

-Is the sample size sufficient to ensure adequate power to address the hypothesis being tested?

-Were correct statistical analysis used to support conclusions?

-Are there concerns about ethical or regulatory requirements being met?

Reviewer #1: The article addresses a relevant topic regarding the connection between risk factors and chikungunya prevalence. However a few details could improve the readability and clarity.

The methodology section provides valuable details on the data sources used in the study.

In line 95, although an encyclopedia [12] can provide general definitions, I suggest substituting it with peer-reviewed scientific sources that empirically describe the socio-environmental conditions of favelas or informal settlements in Brazil, like reference [33]. This would enhance the study's reliability and alignment with the empirical rigor expected in public health research.

To improve clarity, kindly specify in Supplemental Table 1 the category to which each variable belongs.

In line 132, kindly indicate the version of QGIS and the Semi-Automatic Classification Plugin utilized in the analysis.

Additionally, it would be helpful to clarify the rationale behind using a satellite image from a different period than the sampling timeframe. Could the authors provide justification for this choice and include it in the Methods section? Ensuring consistency between the imagery and field data collection period would strengthen the methodological coherence of the study.

Reviewer #2: (No Response)

Reviewer #3: The manuscript by Travis and colleagues aims to identify individual, household, and environmental factors that may contribute to exposure to Chikungunya virus (CHIKV) in four communities in Salvador, a large Brazilian city. To this end, a cross-sectional serological survey was conducted, and blood samples and data were collected from 1318 participants, as well as a measurement of the environmental characteristics of the communities, between March and October 2018.

The authors obtained approval from the ethics committee of the institution where the study was conducted and the National Commission for Research Ethics, Brazilian Ministry of Health; as well as the free and informed consent form of the participants.

My experience with statistical analysis is limited, so my observations and interpretations may not be as in-depth in this regard.

**Results**

-Does the analysis presented match the analysis plan?

-Are the results clearly and completely presented?

-Are the figures (Tables, Images) of sufficient quality for clarity?

Reviewer #1: In Table 1, the authors list the category "Boot access" with its subcategories (No, Own boots, Can borrow boots), whereas the other variables are displayed in a binary Yes/No format. Could you clarify the rationale for using subcategories in this specific case? To ensure consistency, I recommend standardizing all variables as dichotomous.

The order of results shown in lines 207 and 215-216, along with the data layout in Figure 1, lacks a distinct logical structure, potentially complicating interpretation for readers. I suggest rearranging the presentation to highlight essential epidemiological, magnitude, or geographic trends, which will enhance clarity and coherence.

Reviewer #2: (No Response)

Reviewer #3: The manuscript presents interesting data on factors that may increase the risk of exposure to arboviruses in poor communities, such as the degree of urbanization and flood risks. However, much of the data is described in a fragmented way and presented in tables as a whole, making it difficult to understand. Some changes in the way it is presented or described would improve the fluidity of the reading and understanding.

(1) Lines 197-199: “Overall, study participants of all areas were relatively young, with median ages ranging from 26 in RS to 38 in MR, and an overall median age of 33 (IQR 19-48 years).” Where this data is located, which table? Please facilitate identification.

(2) Lines 203-205:”MR had a mean of 49.62m (IQR 44.0-55.4m) above sea level, AC was 54.74m (IQR 45.0-61.76m), RS was 64.65m (IQR 57.54- 73.00), but NC was far lower than other communities at 8.54m (IQR 4.83- 12.21m)”. Where this data is located, which table? Please facilitate identification.

(3) Lines 205-208: “the proportion of participants in each community that reported living in a household on a hillside, with only 3.3% of participants reporting this in NC compared to 27.6%, 20.8% and 59.1% in MR, AC and RS, respectively.” Please, mention where this information is found (supplementary table 2), and in the table include a column with the percentage, to facilitate reading.

**Conclusions**

-Are the conclusions supported by the data presented?

-Are the limitations of analysis clearly described?

-Do the authors discuss how these data can be helpful to advance our understanding of the topic under study?

-Is public health relevance addressed?

Reviewer #1: As a suggestion, the authors could organize the discussion paragraphs in the order of the domains presented: individual, household, and environmental.

The study assumes that transmission occurs primarily within household environments, as suggested by the proximity to environmental risk factors. However, in lines 372-374, the discussion argues that exposure beyond households—such as during daily travel—drives risk in men. This may lead to a potentially ambiguous interpretation.

Additionally, one of the cited studies [10], conducted in a nearby city (Feira de Santana), found a higher risk in women. Could you incorporate a discussion contrasting these findings to provide a broader perspective on possible gender-related differences in exposure?

Reviewer #2: (No Response)

Reviewer #3: The limitation of this study is the use of an anti-CHIKV IgG test to detect seropositivity for CHIKV, since these antibodies indicate a past infection or previous exposure to the virus, and not an acute or recent infection. Furthermore, the presence of IgG can persist for years, making it difficult to distinguish between recent and past infection. In this case, the most appropriate technique would be the viral RNA test by RT-PCR (Reverse Transcription Polymerase Chain Reaction). This test can identify the genetic material of the virus during the first few days of infection, usually up to about a week after symptoms begin. IgM antibodies could also be used to detect recent infections, as they appear soon after the acute phase.

**Editorial and Data Presentation Modifications?**

Reviewer #1: As a minor revision that does not substantially alter the conclusions, I suggest improving consistency in Table 1. The authors list the category "Boot access" with subcategories (No, Own boots, Can borrow boots), while the other variables are presented in a binary Yes/No format. Could you clarify the rationale for using subcategories in this specific case? To ensure uniformity, I recommend standardizing all variables as dichotomous.

Reviewer #2: (No Response)

Reviewer #3: Suggestions:

Clarify in the methodology or at the beginning of the results that “of the 1318 participants who provided a blood sample, 1316 had conclusive serological results and that not all answered all the questions”, to clarify the inconsistency in the total number of participants (%) in Table 1, and why the description of the result says 1,318 and the title of the table says 1,316.

When talking about the collection period (between March and October 2018), explain the importance/impact of this period in increasing exposure to CHIKV. Example: is it a rainy season in this region? Does the climate during this period favor the proliferation of the vector and consequently exposure to CHIKV?

**Summary and General Comments**

Reviewer #1: The research provides valuable insights from a multidisciplinary and cross-sectional perspective. Its strengths lie in recognizing that environmental factors have a more significant impact on risk than individual-related factors. The findings highlight the crucial role of public management in overseeing urban areas, particularly informal settlements that house marginalized communities vulnerable to CHIKV

Although these are widely recognized concepts, the authors may consider citing studies that characterize the structural deficiencies in informal settlements and how these features influence vector ecology, particularly in the sentence starting from lines 61 to 64. There are a few references used in the discussion that could be used here.

The sentence starting at lines 76-80 appears to have excessive repetition of the term 'environmental features.' I would recommend revising it to improve readability. Additionally, it would be helpful to standardize the hyphenation of the term 'fine-scale' found in lines 72 and 82.

In line 86, I suggest reviewing whether the term 'microenvironment' is the most appropriate to describe the habitat surrounding the households.

Reviewer #2: (No Response)

Reviewer #3: The study presents findings that reinforce the association between a higher number of breeding sites and seropositivity and environmental deficiencies such as lack of water coverage and risk of flooding. It also addresses important concerns about how environmental deficiencies and inadequate housing can increase the risk of exposure to arboviruses. In addition, it presents relevant data that can be presented to local authorities to take measures to improve sanitation and housing conditions in the region.

On the other hand, some results could be presented more clearly to facilitate reading and understanding. The study also acknowledges its limitations, which is important, but these do not completely invalidate the findings. Nevertheless, they suggest that future research could delve deeper into these issues, taking these points into account.

PLOS authors have the option to publish the peer review history of their article (what does this mean? ). If published, this will include your full peer review and any attached files.

**Do you want your identity to be public for this peer review?** For information about this choice, including consent withdrawal, please see our Privacy Policy .

Reviewer #1: **Yes: ** Natan Diego Alves de Freitas

Reviewer #2: No

Reviewer #3: No

**Figure resubmission:**

**Reproducibility:**



---

## [Decision Letter · Decision Letter 1]

14 Aug 2025

Dear Prof Costa,

We are pleased to inform you that your manuscript 'Topography and environmental deficiencies are associated with chikungunya virus exposure in urban informal settlements in Salvador, Brazil' has been provisionally accepted for publication in PLOS Neglected Tropical Diseases.

Best regards,

Michael R Holbrook, PhD

Section Editor

Elvina Viennet

Section Editor

Shaden Kamhawi

co-Editor-in-Chief

Paul Brindley

co-Editor-in-Chief

Reviewer's Responses to Questions

**Key Review Criteria Required for Acceptance?**

**Methods**

-Are the objectives of the study clearly articulated with a clear testable hypothesis stated?

-Is the study design appropriate to address the stated objectives?

-Is the population clearly described and appropriate for the hypothesis being tested?

-Is the sample size sufficient to ensure adequate power to address the hypothesis being tested?

-Were correct statistical analysis used to support conclusions?

-Are there concerns about ethical or regulatory requirements being met?

Reviewer #1: The article addresses a relevant topic regarding the connection between risk factors and chikungunya seroprevalence. The methodology section provides valuable details on the data sources used in the study. The study design, which combines household surveys, environmental variables, and spatial analysis, appears appropriate to address the objectives. The population is clearly described, and the focus on a vulnerable area affected by a chikungunya outbreak is suitable for investigating socio-environmental risk factors.

Reviewer #3: Travis and colleagues aimed to identify factors that may contribute to exposure to the Chikungunya virus (CHIKV). To achieve this, they collected individual and household data, as well as blood samples from 1,318 individuals who consented to participate in the study. Environmental data were also gathered from the communities involved. Additionally, the authors obtained approval from the local ethics committee.

**Results**

-Does the analysis presented match the analysis plan?

-Are the results clearly and completely presented?

-Are the figures (Tables, Images) of sufficient quality for clarity?

Reviewer #1: The results presented is consistent with the analysis plan described in the Methods. The authors apply appropriate statistical approaches to address clustering at the household level and explore associations between socio-environmental variables and chikungunya seropositivity. The rationale behind variable selection and model construction is well-described and aligns with the stated objectives. Tables and figures are informative and relevant to the hypotheses tested.

Reviewer #3: The article provides valuable information about factors that may increase the likelihood of contact with CHIKV in vulnerable communities, such as urbanization and flood risk. The findings are presented clearly and transparently, making them easy to understand and enhancing the overall reading experience.

**Conclusions**

-Are the conclusions supported by the data presented?

-Are the limitations of analysis clearly described?

-Do the authors discuss how these data can be helpful to advance our understanding of the topic under study?

-Is public health relevance addressed?

Reviewer #1: The revised conclusion has become more succinct and clearer, which improves overall readability and strengthens the take-home message of the study. The inclusion and specification of variable domains also contribute to a better understanding of how different risk factors were categorized and interpreted, supporting a more structured and coherent synthesis of the findings.

Reviewer #3: The conclusion is clear and well-articulated, effectively summarizing the main findings of the study. The limitations are also explained, indicating that future research could explore these aspects in more depth and take them into consideration.

**Editorial and Data Presentation Modifications?**

Reviewer #1: Given the importance of the topic and the quality of the work, I recommend the article be accepted.

Reviewer #3: The data are presented clearly and completely.

**Summary and General Comments**

Reviewer #1: The research provides valuable insights from a multidisciplinary and cross-sectional perspective. Its strengths lie in recognizing that environmenta factors have a more significant impact on risk than individual-related factors. The findings highlight the crucial role of public management in overseeing urban areas, particularly informal settlements that house marginalized communities vulnerable to CHIKV.

Reviewer #3: The manuscript reinforces the connection between increased exposure to arbovirus and environmental deficiencies, providing valuable data that can be shared with local authorities to support the implementation of measures aimed at improving sanitation and housing conditions in the region.

PLOS authors have the option to publish the peer review history of their article (what does this mean? ). If published, this will include your full peer review and any attached files.

**Do you want your identity to be public for this peer review?** For information about this choice, including consent withdrawal, please see our Privacy Policy .

Reviewer #1: No

Reviewer #3: No

---

## [Editor Report · Acceptance letter]

Dear Prof Costa,

We are delighted to inform you that your manuscript, " 

Topography and environmental deficiencies are associated with chikungunya virus exposure in urban informal settlements in Salvador, Brazil," has been formally accepted for publication in PLOS Neglected Tropical Diseases.

Best regards,

Shaden Kamhawi

co-Editor-in-Chief

Paul Brindley

co-Editor-in-Chief
